# Facile Synthesis of ZIF-67 for the Adsorption of Methyl Green from Wastewater: Integrating Molecular Models and Experimental Evidence to Comprehend the Removal Mechanism

**DOI:** 10.3390/molecules27238385

**Published:** 2022-12-01

**Authors:** Muniba Ikram, Sadaf Mutahir, Muhammad Humayun, Muhammad Asim Khan, Jehan Y. Al-Humaidi, Moamen S. Refat, Amr S. Abouzied

**Affiliations:** 1Department of Chemistry, University of Sialkot, Sialkot 51300, Pakistan; 2School of Chemistry and Chemical Engineering, Linyi University, Linyi 276000, China; 3Wuhan National Laboratory for Optoelectronics, School of Optical and Electronics Information, Huazhong University of Science and Technology, Wuhan 430074, China; 4Department of Chemistry, College of Science, Princess Nourah bint Abdulrahman University, P.O. Box 84428, Riyadh 11671, Saudi Arabia; 5Department of Chemistry, College of Science, Taif University, P.O. Box 11099, Taif 21944, Saudi Arabia; 6Department of Pharmaceutical Chemistry, College of Pharmacy, University of Hail, Hail 81442, Saudi Arabia; 7Department of Pharmaceutical Chemistry, National Organization for Drug Control and Research (NODCAR), Giza 12553, Egypt

**Keywords:** zeolitic imidazolate framework (ZIF), methyl green, adsorption, density functional theory, MD simulations, water pollution

## Abstract

Organic dyes with enduring colors which are malodorous are a significant source of environmental deterioration due to their virulent effects on aquatic life and lethal carcinogenic effects on living organisms. In this study, the adsorption of methyl green (MG), a cationic dye, was achieved by using ZIF-67, which has been deemed an effective adsorbent for the removal of contaminants from wastewater. The characterization of ZIF-67 was done by FTIR, XRD, and SEM analysis. The adsorption mechanism and characteristics were investigated with the help of control batch experiments and theoretical studies. The systematical kinetic studies and isotherms were sanctioned with a pseudo-second-order model and a Langmuir model (R^2^ = 0.9951), confirming the chemisorption and monolayer interaction process, respectively. The maximum removal capacities of ZIF-67 for MG was 96% at pH = 11 and T = 25 °C. DFT calculations were done to predict the active sites in MG by molecular electrostatic potential (MEP). Furthermore, both Molecular dynamics and Monte Carlo simulations were also used to study the adsorption mechanism.

## 1. Introduction

Several organisms, including humans, cannot survive without a steady supply of clean water [1]. Manufacturing operations released water into the environment despite its contamination with heavy metal ions, dangerous dyes, toxic pigments, and noxious organic solvents [2]. It is a serious risk to humans and other organisms because it pollutes both surface and groundwater. The textile, paper, tanning, cosmetic, and pharmaceutical sectors all produce organic dye wastewater [3], which is very hazardous and cannot be decomposed into the environment [4]. In order to ensure the continued existence of aquatic life, toxic compounds must be removed from the environment [5,6].

MG is a harmful substance that has widespread industrial applications including leather, plastic, textile, paper, dyeing, and printing [7]. Consequently, decontamination of water containing MG before releasing it into the ecosystem is both a moral and practical imperative [8]. Biodegradation, photo-degradation, adsorption, and chemical oxidation are only a few of the methods that have been studied and published to remove harmful contaminants from water [9]. In comparison to the other methods we have examined, adsorption is the most beneficial and commonly accepted because of its low cost, energy efficiency, lack of secondary pollution, and ease of operation [10].

Therefore, chemists have continuously sought out novel materials exhibiting extraordinarily high adsorption efficiency. Bentonite, fly ash, silica, carbon nanotubes, activated carbon, charcoal, biochar, clay, agricultural wastes, industrial solid wastes, and grafted polymer are only a few of the many materials recorded in the literature as being effective at removing dangerous chemicals from water [11,12,13,14,15].

Recently, Metal-Organic Frameworks (MOFs) have evolved as an emerging class of functional materials for a variety of applications due to their high porosity, high surface area, tunable internal structure, and flexible design [16]. MOFs are formed when metal ions or clusters form covalent bonds with organic ligands [17]. Additionally, the surface of the functionalized MOF materials might be altered to have more active sites so that they can interact with the guest molecule through π-π, hydrogen bonding, and electrostatic interactions. This makes MOFs an attractive choice for efficient adsorption procedures. There exists a significant subclass of metal-organic frameworks (MOFs) known as zeolitic imidazolate frameworks (ZIFs) [18], which are composed of metal centers (such as Co, Zn, Fe, etc.) and imidazolate linkers [19]. Due to their microporosity, high thermal and chemical stability, and higher surface area, ZIFs are quickly replacing traditional adsorbent materials [20].

Herein, we espoused a simple and cost-effective route for the synthesis of ZIF-67 and explored its potential to remove MG by adsorption. To the best of our knowledge, no research has been published yet about the removal of MG through ZIF-67. To identify chemical and physical features, the produced substance was evaluated using several analytical methods. The batch adsorption process was approximated by adjusting the experimental parameters, such as adsorbent dosage, MG concentration, pH, and duration. Langumir and Freundlich models were used to simulate the adsorption isotherm data. The interactions between the MG molecule and the ZIF-67 surface were finally explained using computational modelling and kinetic models. This work demonstrates the potential of ZIF-67 as an effective adsorbent for MG adsorption (Figure 1).

## 2. Experimental

### 2.1. Materials

The 2-Methyl Imidazole 99% (C_4_H_6_N_2_), Cobalt (II) nitrate hexahydrate 98%, [Co(NO_3_)_3_·6H_2_O], Methanol 99.8% (CH_3_OH), Ethanol 98% (C_2_H_5_OH), Sodium hydroxide (NaOH) and Hydrochloric acid (HCl) were received from Sigma-Aldrich. The entire experiment utilized laboratory-prepared distilled water. All starting ingredients, reagents, and solvents were utilized with no additional purification.

### 2.2. Synthesis of Materials

ZIF-67 was prepared by a previously reported approach with just modest alterations [21]. Solution 1 was made by dissolving 2-Methylimidazole (0.8 M) in methanol (50 mL), while solution 2 was made by dissolving cobalt nitrate (0.1 M) in methanol (50 mL). When the two liquids were shaken vigorously, the chemicals were dispersed. Drop by drop, the solutions B and A were blended and agitated for an hour at 25 °C. The mixture was then kept on stirring for 24 h and finally, ZIF-67 was taken out of a combination by centrifuging it at 6000 revolutions per minute for 20 min, washing it three times with water and methanol, and finally drying it at 50 °C in the vacuum oven for 24 h.

### 2.3. Characterization

X-ray Diffractometer (HZG41B-PC, Victoria, Australia) was utilized to obtain XRD patterns of manufactured ZIF-67. The scanning Electron Microscopy (SEM) (JEOL JSM 5910, Tokyo, Japan) technique was performed to investigate the structure and surface topology. The Shimadzu IR Presige-21 spectrophotometer was used to obtain FT-IR (IRSpirit-T, Shimedzu, Kyoto, Japan) spectra of produced ZIF-67 by employing KBr pellets. UV-Vis spectrophotometer (Cecil 7400S) was used to collect adsorption data of dye Methyl Green (MG) before and after adsorption.

### 2.4. Batch Adsorption Experiment

In a 100 mL Erlenmeyer flask, experiments were run in batch mode and shaken for 120 min at 120 rpm in an orbital shaker (orbital shaker incubator ES 20). The original pH was adjusted using HCl and NaOH. By conducting MG adsorption at various starting concentrations (10–50 mg/L), adsorbent dosages (0.5–2.5 g), times (20–140 min), and pH ranges (2–12), the effects of various factors were examined. Each experiment was carried out three times to ensure accuracy and repeatability. A Cecil 7400S UV-Vis spectrophotometer was used to measure the amounts of MG at the maximal absorption wavelength, max = 592 nm. Equations (1) and (2) were used to obtain the adsorption capacity q_e_ (mg/g) and removal effectiveness (%), respectively.
(1)$$q_{e}=\frac{\left(C_{o}-\mathrm{Ct}\right)V}{W}$$
(2)$$\%=\frac{C_{o}-C_{e}}{C_{o}}$$

Here, C_e_ and C_o_ stand for the initial and equilibrium concentrations of the adsorbate (mg/g), respectively, while V denotes the volume of solution (L) and W denotes the weight of the adsorbent (g) utilized.

### 2.5. Adsorption Kinetics

Kinetic models are sought out to describe adsorption features and adsorption mechanisms. Pseudo-first order and pseudo-second order models are used to describe the adsorption process. The concentration of the solution was measured at various time intervals, and the adsorption capacity at that point is specified as qt and derived by the subsequent equation.

Pseudo-first order is given by Equation (3)
(3)$$\ln^{`}\left(\frac{q_{e}}{q_{e-}q_{t}}\right)=\frac{K_{1}t}{2.303}$$

While q_t_ and q_e_ show adsorption capacities at time t and equilibrium, respectively, K_1_ (1/min) exhibits an adsorption rate constant for pseudo-first order.

Equation (4) provides a kinetic model of pseudo-second order.
(4)$$\frac{t}{q_{t}}=\frac{t}{q_{e}}+\frac{1}{K_{2}q_{e}^{2}}$$

K_2_ is a pseudo-second-order constant, while q_t_ (mg/g) and q_e_ (mg/g) are the adsorption capacities at time t and equilibrium, respectively.

### 2.6. Adsorption Isotherm

Adsorption isotherms show how adsorbents and adsorbates interact. The quantity of adsorbate adsorbed on the adsorbent and the amount present in the solution at equilibrium are shown in relation to one another. These models may also be used to deduce the adsorption mechanism. The Freundlich and Langmuir adsorption isothermal models have been used in this study to assess adsorption equilibrium data. Langmuir’s model is based on the homogeneous surface of the adsorbent and the monolayer adsorption process [22]. The following equation can be used to obtain the Langmuir isotherm.
(5)$$\frac{C_{e}}{Q_{e}}=\frac{1}{K_{L}q_{m}}+\frac{1}{q_{m}}C_{e}$$

Here, C_e_ stands for equilibrium concentration (mg/L), Q_e_ for equilibrium adsorption capacity (mg/g), and K_L_ for the Langmuir constant, while q_m_ is the maximum adsorption capacity determined by the Langmuir isotherm. The slope and intercept of the fitted line of the Langmuir isotherm are used to compute the data acquired by the Langmuir isotherm.

The Freundlich isotherm is predicated on the idea that a multi-layer adsorption process is responsible for adsorption on the heterogeneous surface [23]. The following equation yields the Freundlich isotherm.
(6)$${\mathrm{lnq}}_{e}=\frac{1}{n}{\mathrm{lnC}}_{e}+{\mathrm{lnK}}_{F}$$

K_F_ stands for the Freundlich constant (L/mg), n for adsorption strength, and 1/n for adsorption favorability, which is based on the degree of surface heterogeneity [24].

### 2.7. Effect of Different Parameters

The adsorption process may be drastically altered if the solution’s pH is altered. We adjusted the pH of the solution by adding either HCl or NaOH. Next, 20 mL of the prepared stock solution was collected in four separate flasks, while 2.0 g/L g ZIF-67 was added in all the flasks. HCl and NaOH were employed to alter the pH at from 2 to 12. The flasks were put in the orbital shaker for 120 min at 150 rpm. After 120 min, all the flasks were disconnected from the shaker and filtered. UV-Vis spectrophotometer was utilized to measure the adsorption and 20 mL of the prepared stock solution was collected in four different flasks. Next, 2.0 g/L of ZIF-67 was added in all the flasks. The flasks were placed in a shaker for different time intervals including 20, 40, 60, 80, 100, 120, and 140 min at 150 rpm. After different time intervals, flasks were detached from the shaker and filtered and 20 mL of the prepared stock solution was collected in four different flasks. Different concentration of ZIF-67 ranging from 0.5–2.5 g/L was added to the flasks. The flasks were placed in the orbital shaker at 150 rpm in a shaker. After 120 min, the flasks were detached from the shaker and filtered for adsorption efficiency calculations [25].

## 3. Results

A Scanning Electron Microscope (SEM) was used to study the sample’s structure. The surface topography of synthesized material was examined at high magnification using a focused electron beam in SEM. Figure 1a,b depicted SEM images of highly porous structures. No difference in SEM pictures compared to prior approaches was found [26]. The specific morphologies likely lead to customizable structures and the ability to adsorb organic dyes. ZIF-67 crystals displayed polyhedral shapes with relatively narrow size distribution.

A fundamental use of FT-IR is to determine the functional group of a molecule. The functional groups of ZIF-67 were studied using an FT-IR spectrometer (Figure 1c). The spectra depicted ZIF-67 peaks ranging from 375 to 1750 cm^−1^. Peaks in the ZIF-67 spectra at 1303 cm^−1^ and 1420 cm^−1^ reflected ring-strengthening vibrations of imidazole and out-of-plane whirling and twisting vibrations of CH_2_, the ZIF-67 FT-IR spectra are consistent with the literature [27]. Stretching vibrations of alkenes (-C=C-) peak at 1640 cm^−1^. Whereas C-O stretching vibrations peak at 1561 cm^−1^ for –COOR groupings. The crystallinity of ZIF 67 was determined using XRD JDX0 362 (Japan) with Cu Kα radiation (wavelength 1.5418) with a scan angle range of 5° to 90° with a 40 kV acceleration voltage and 30 mA current. Figure 1d depicts the patterns of ZIF-67 which are in accordance with the simulated pattern [28]. From all of the above characterization, we found that the product was pure phase ZIF-67 materials.

## 4. Adsorption Experiments

A UV–Visible spectrophotometer (SPECORD 210 PLUS-223F1719C) was used to scan the spectrums of varied MG concentrations [29]. Adsorption was tracked using spectrophotometry after samples were centrifuged at 6000 rpm for 15 min to remove residue after a predetermined contact period. MG’s basic absorption peak occurs at a wavelength of max = 618 nm, hence this is the wavelength at which the adsorption experiment was conducted using a UV-Vis spectrophotometer (Figure 2) [27].

### 4.1. Effect of pH

The pH of the solution is a crucial element in affecting the efficiency of an adsorbent. It has the potential to alter the charge of dye molecules and adsorbents as well as ionize various pollutants, all of which can lead to dissociation at the active sites. The weighed amount of ZIF-67 was added to the dye solution of different pH. The pH of the initial dye solutions varied from pH 11. Each sample was kept in a rotary shaker for 120 min. The concentration of dye in supernatant liquid was determined after filtration by UV-vis spectrophotometer. The results depicted in Figure 3a, indicate that maximum removal of MG was obtained at pH 11. According to the results, the removal efficiency increases from 65–98%, indicating that the alkaline medium is favored for MG removal by adsorption process. It is because at a low pH, there is an excess of H^+^ ions concentration which built a competition between H^+^ ions and dye to interact with the adsorbent and being small in size as compared to dye, H^+^ ions occupy active sites more quickly [30].

### 4.2. Effect of Contact Time and MG Concentration

Dye removal was studied in relation to contact time and MG concentration at the outset. The results of removing MG at pH 11 with ZIF-67 are shown as a percentage over time in Figure 3b,d. It was discovered that after the first 50 min, MG adsorption onto ZIF-67 slowed down to the point where over 60% of the MG concentration was removed. The rapid uptake of MG at low contact times can be attributed to the ZIF-67’s highly negatively charged surface for the adsorption of MG in the solution at pH 11. For relatively high dye concentrations and low adsorbent masses 2.0 g/L, the removal efficiency was critical, demonstrating the usefulness of the economically active material in the complete removal of cationic dye while the slow rate of MG adsorption is likely due to electrostatic hindrance or repulsion between the adsorbed, negatively charged adsorbate species onto the surface and the available adsorbate species in the solution, as well as the slow pore diffusion of the solute ions into the bulk of the adsorbent. The equilibrium time was determined to be 120 min after maximum MG adsorption onto ZIF-67 was achieved [31].

### 4.3. Effect of Initial Adsorbate Concentration

ZIF 67 (2.0 g/L) at pH 11 was used to remove MG from solutions of varying initial concentrations. The relationship between the concentration of MG in the solution and the adsorption capacity of ZIF 67 was also studied (Figure 3c). The test solution was conducted at room temperature (27 °C), pH 11, with varying initial concentrations of ZIF-67 (0.5 to 2.5 g L^−1^). This shows that the adsorption efficiency of ZIF 67 increased from 0.5 to 2.0 g/L (MG concentration 100 mg/L) and then decreased when adsorbate concentration was increased to 2.5 g/L in the starting solution [32].

### 4.4. Adsorption Kinetics

ZIF 67’s ability to adsorb MG was examined by using first and second-order models. The pseudo-first-order kinetic model makes it easy to understand the link between the adsorbent capacity and dye adsorption rate [33]. First-order kinetic models are widely employed in the early stages of adsorption. A pseudo-second-order kinetic model is used to estimate the chemisorption mechanism as it moves toward equilibrium [34]. Pseudo-second order has a better match than the pseudo-first order (R^2^ = 0.91875). ZIF 67’s MG removal was dominated by chemisorption [35], as shown more accurately by the pseudo-second order model of the adsorption process than by other models (Figure 4a,b).

### 4.5. Adsorption Isotherms

ZIF-67’s adsorption capabilities were studied by testing different concentrations of MG, starting at 10 mg·L^−1^ 1 to 50 mg·L^−1^ raising the MG content results in a greater adsorption capacity across the board so that the equilibrium features of MG adhesion to ZIF-67 may be further explained. Experimental data was simulated by utilizing the Freundlich and Langmuir isotherms models [36].

According to the Langmuir model, no further adsorption may occur when the uniform surface of an adsorbent material is occupied by adsorbate molecules [37]. Including that the adsorbent surface is not smooth and that the adsorption process takes place in multiple stacked levels. This model’s R^2^ coefficient was 0.996 for the Langmuir isotherm, indicating that the Langmuir model better described the process (Figure 4c,d) [27].

### 4.6. Reusability of ZIF-67

The applicability of any selected adsorbent used on a commercial scale greatly depends on its ability to be recycled after adsorption. The reusability of the synthesized ZIF-67 was determined by reutilizing it up to four times. The most promising conditions such as concentrations of MG (100 mg/L), pH (11), adsorbent dose (2.0 g/L), and contact time (120 min) kept constant for each cycle. After each adsorption, ZIF-67 was washed with 50% ethanol and vacuum dried at 50 °C for 24 h. The adsorption capacity of the adsorbent declined slightly with increasing cycles but the value was still substantial. This suggests that ZIF-67 is stable and can be reused for the adsorptive removal of MG.

### 4.7. Examining the Adsorption Mechanism Using DFT Analysis

Gaussian software was used to do the DFT computations and geometry optimizations of ZIF-67 and MG (Figure 5a,b). Becke’s three-parameter hybrid functional (B3LYP) with the LYP correlation functional was employed in all computations. It reproduces the geometries of both small and big molecules. It is commonly employed in thermodynamic simulations of organic and organometallic systems. The 6-31G basis set was chosen as a balance between calculation quality and computational expense (for optimizations Calculation).

To study adsorption mechanisms and forecast reactive locations for electrostatic interactions, theory calculations were used. DFT calculations were carried out by utilizing a basis set of 3-21G B3YLP levels to look into potential molecular interactions between ZIF-67 and MG. The optimized shape of the MG molecule is depicted in Figure 5a. The HOMO and LUMO of ZIF-67 (Figure 5b) and MG calculations were used to investigate the inter-action mechanism (Figure 5c). The calculated energy gap between HOMO and LUMO for ZIF-67 is 2.59 eV, whereas the calculated energy gap for MG is 7.121 eV. This difference indicates that MG requires more energy for the transition from LUMO to HOMO, whereas ZIF-67 requires less energy, indicating that it is more reactive. Their parameters are shown in Table 1.

The sections of a molecule with greater and lower potential are shown by the colors blue, yellow, and red in the MEP (Molecular Electrostatic Potential) study. Figure 5d,e shows MEP plots of ZIF-67 and MG, indicating that the N atom of the imidazole ring is at a lower potential, which is shown by its reddish-yellow color, and may be a potential target for electrophilic assault. The H atoms in the imidazole ring have a lower potential, giving them a blue hue, and they may be vulnerable to nucleophilic assault. According to the MEP plot, the nitrogen of MG will experience nucleophilic contact, while the side chains of MG will experience electrophilic interaction. The results of the DFT study also support that due to the electrostatic interaction between the ZIF-67 and MG the adsorption process is spontaneous [38].

Chemical potential (µ), electronegativity (χ), hardness (η), softness (S), and the electrophilicity index (ω) were generated from HOMO and LUMO energies to describe the global chemical reactivity of materials (Table 1).

### 4.8. Adsorption Mechanism

The molecular level adsorption process of methyl green (MG) by ZIF-67 was investigated using Monte Carlo (MC) and Molecular dynamic (MD) simulations. Frenkel and Smit have provided a description of the fundamental ideas behind MC and MD simulations [39]. The MG desorption sites on the ZIF-67 were discovered using the MC simulation. The MC lowest-energy structures were simulated in explicit water using MD to evaluate the impact of the presence of solvent molecules (water) on the MG adsorption.

Figure 6a depicts MG’s adsorption on the surface of ZIF-67 in a dry solution (no solvent). Due to the fact that the MG molecule includes many HB donor and acceptor sites, it has created a number of hydrogen bonds with the hydroxide or –NH groups on the ZIF-67 surface. Together with the hydroxyl hydrogen atoms on the ZIF-67 surface, the amino group’s oxygen and nitrogen atoms created HBs. In MG, hydroxyl hydrogen atoms with keto groups, amine nitrogen atoms, and hydroxyl oxygen atoms were also used to generate intramolecular HBs.

As shown in Figure 6b, the two-layer model was used to study the adsorption of MG rather than the ZIF-67 surface model in order to determine if the MG can bind to the metal ions (Zn^2+^) of the adsorbent. Different free bonds exist among the metal ions in the two layers model. Metal ions with various free bonds can be found on each edge of a single sheet. For instance, a metal ion-containing edge has no free bonds, a metal ion-containing edge has two free bonds, and a metal ion-containing edge has three free bonds. In MG molecules, the functional groups were also seen to create intramolecular hydrogen connections with one another.

Figure 6c depicts MD snapshots for the MG adsorption on the ZIF-67 surface and two-layer models when there is water present. The HO-Zn groups and the keto and hydroxyl groups of the MG molecule generated HBs in the event of MG adsorption on the ZIF-67 surface in water (Figure 6c). The tertiary and primary amine groups do not combine with the ZIF-67 surface to create HBs. The MG molecule established coordination interactions with Zn^2+^ atoms in water, as shown in Figure 6d. Both intramolecular HBs between the functional groups of the MG molecule and HBs between MG and water molecules were seen in both water systems. Consequently, the MD simulation demonstrates that MG interacts with the ZIF-67 even when water molecules are present. A comparison study has been presented in Table 2 for the adsorption capacity of various MOFs-based materials against MG removal.

## 5. Conclusions

In conclusion, ZIF-67 was successfully synthesized by a simple cost-effective method and ascertained as an excellent absorbent for the removal of MG from wastewater. The results revealed that the synthesized ZIF-67 performed best in a basic medium (pH = 11). ZIF-67 is made from organic carboxylates and imidazoles such as 1,2 methyl imidazole linked to multimetal clusters. ZIF-67 has a sturdy structure with permanent porosity, a cost-effective and ecologically beneficial material. ZIF-67 highlight the extraordinary flexibility and thermal stability of metal-ligand interactions in producing new compounds with fascinating shapes and characteristics. The most remarkable aspects of ZIF-67 are their enormous specific surface areas and tunable nanoscale porosity, which are permanent, with evenly ordered cavities. The facile synthesis of ZIF-67 was carried out in an easy manner, followed by alteration of their porosity and increased surface area, leading to exceptional results in the removal of MG from water. The produced ZIFs were characterized by FT-IR, SEM, and XRD. It was successful to use the first- and second-order pseudo-models, respectively. At pH 6, complete decolorization of MG was obtained. R^2^ value 0.99081 depicted that the adsorption process followed Langumir and Freundlich adsorption isotherm. R^2^ value 0.9951 depicted that the adsorption process followed the Langumir adsorption isotherm. Electrostatic attraction, π-π interaction, and complexation were the primary mechanisms for pollutant absorption. As a starting point for further investigation, this work employed DFT to determine the specific physical parameters of ZIF-67 and MG. Ultimately, this suggested effort will have ramifications for the adsorption of MG by ZIF-67 as it is economical and worthwhile.

## Data Availability

Not applicable.

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
