# Peer review of "Facile Synthesis of ZIF-67 for the Adsorption of Methyl Green from Wastewater: Integrating Molecular Models and Experimental Evidence to Comprehend the Removal Mechanism"

_molecules, 2022, doi:10.3390/molecules27238385_

Round 1

Reviewer 1 Report

MDPI: molecules-1999819

Title: Facile synthesis of ZIF-67 for the adsorption of methyl green from wastewaters: integrating molecular models and experimental evidence to comprehend the removal mechanism

I would not recommend the submitted manuscript for the following comment:

Abstract:

1.       Line 23: The authors should not use abbreviation with addressed it in the abstract.

2.       The authors should raise significant outcome of the research study in the abstract. I cannot find out the result of mechanisms, kinetic and even performance of the research been done I the abstract. To be in mind, abstract is very important to raise the attention of such manuscript among the reader. Poor write up of abstract.

Introduction:

1.       The authors should rewrite the last paragraph of manuscript by stated the objectives of the research study rather than stated the result of research summarized methodology. Reviewer feel that is not suitable write as introduction of methodology ways.

2.       The authors should state the gap analysis of the proposed study from the current literature about the intermolar and kinetics are not included in the introduction.

Methodology:

1.       Please check the guideline of the MDPI. I never seen that research methodology is addressed after the result. It should be front before the result and discussion section.

2.       The research methodology not addressed completed by the authors. Where is the adsorption experimental design? Result and discussion stated performance; kinetics and intermolar reaction, but I cannot find out how does the experimental done?  

Result and discussion

1.       Line 126: Where is the figure 2 as stated by the authors?

2.       Figure 4 please included the R2 int the graph and stated clear the model. Therefore, table 1 and 2 are not necessary included in the manuscript.

3.       Detail discussion should be included on the model kinetics obtained with comparison the authors work with others works.

4.       DFT Studies; Molecular Electrostatic Potential Surface and Global Reactivity Descriptors are the characteristic about the material ZIF-67. The authors wrote mostly here about the literature of the material rather then the result finding from the experiment. These 3 sections totally are not related to the research proposed. I cannot find the flow of manuscript of interconnection this section with the result finding.

5.       Poor format checking. Line 206: Table 2…since line 152 already stated it before.

Overall, I would suggest major correction for the submitted manuscript.

Author Response

Dear and respected,

Ms. Claire Gao

Managing Editor,

Molecules.

Manuscript Ref. No.: MDPI: molecules-1999819

Article Type: Full-length article

Article Title: Facile synthesis of ZIF-67 for the adsorption of methyl green from wastewater: integrating molecular models and experi-mental evidence to comprehend the removal mechanism

Thank you very much for letting us revise our manuscript. No doubt the reviewer’s comments are valuable and very helpful for improving our research paper. We have studied the comments carefully and have tried our best to revise the manuscript, which we hope to meet with acceptance requirements. The reviewer’s comments have been yellow highlighted in the manuscript.

REVIEWER'S COMMENTS AND RESPONSES.

Reviewer #1:

Abstract:

  1. Line 23: The authors should not use abbreviation with addressed it in the abstract.

Response: Respected reviewer thank you very much for your valuable comments, abbreviated forms are explained when they appear at first and later on abbreviated forms are written only.

  1. The authors should raise significant outcome of the research study in the abstract. I cannot find out the result of mechanisms, kinetic and even performance of the research been done I the abstract. To be in mind, abstract is very important to raise the attention of such manuscript among the reader. Poor write up of abstract.

Introduction:

  1. The authors should rewrite the last paragraph of manuscript by stated the objectives of the research study rather than stated the result of research summarized methodology. Reviewer feel that is not suitable write as introduction of methodology ways.

Response: Respected reviewer thank you very much for your valuable comments, we have modified and improved last paragraph of the introduction section.

Here in, we espoused simple and cost effective route for the synthesis of ZIF-67 and explored its potential to remove MG by adsorption. To the best of our knowledge no research has been published yet about the removal of MG through ZIF-67. To identify, chemical and physical features, the produced substance was evaluated using several analytical methods. The batch adsorption process was approximated by adjusting the experimental parameters, such as adsorbent dosage, MG concentration, pH, and duration. Langumir and Freundlich models were used to simulate the adsorption isotherm data. The interactions between the MG molecule and the ZIF-67 surface were finally explained using computational modelling and kinetic models. This work demonstrates the potential of ZIF-67 as an effective adsorbent for MG adsorption.

  1. The authors should state the gap analysis of the proposed study from the current literature about the intermolar and kinetics are not included in the introduction.

 Response: Respected reviewer thank you very much for your valuable comments, we have modified and improved the introduction section. We have added a paragraph about intermolecular force and kinetics of adsorption in introduction section.

Methodology:

  1. Please check the guideline of the MDPI. I never seen that research methodology is addressed after the result. It should be front before the result and discussion section.

Response: Respected reviewer thank you very much for your valuable comments, we have moved experimental section before results and discussion section.

  1. The research methodology not addressed completed by the authors. Where is the adsorption experimental design? Result and discussion stated performance; kinetics and intermolar reaction, but I cannot find out how does the experimental done?  

 Response: Respected reviewer thank you very much for your valuable comments, we have moved experimental section before results and discussion section. We have added details about the batch experiments, Adsorption Kinetics, Adsorption isotherm in experimental section.

Result and discussion

  1. Line 126: Where is the figure 2 as stated by the authors?

Response: Respected reviewer thank you very much for your valuable comments, figure 2 has been presented at page 6 of the manuscript. We have also updated its caption for better understanding.

  1. Figure 4 please included the R2 int the graph and stated clear the model. Therefore, table 1 and 2 are not necessary included in the manuscript.

Response: Respected reviewer thank you very much for your valuable comments, we have added “R2” and “qe” values in figure 4 and removed table 1 and 2.

  1. Detail discussion should be included on the model kinetics obtained with comparison the authors work with others works.

Response: Respected reviewer thank you very much for your valuable comments, we have described kinetics data and added references 34, 35 and 36 for comparison with other published work.

  1. DFT Studies; Molecular Electrostatic Potential Surface and Global Reactivity Descriptors are the characteristic about the material ZIF-67. The authors wrote mostly here about the literature of the material rather then the result finding from the experiment. These 3 sections totally are not related to the research proposed. I cannot find the flow of manuscript of interconnection this section with the result finding.

Response: Respected reviewer thank you very much for your valuable comments, we have updated DFT section of the manuscript and made it to the point.

To study adsorption mechanisms and forecast reactive locations for electrostatic interactions, theory calculations were used. DFT calculations were carried out utilising a basis set of 3-21G B3YLP level to look into potential molecular interactions between ZIF-67 and MG. The optimised shape of the MG molecule is depicted in Figure 5(a). The HOMO and LUMO of ZIF-67 (fig. 5b) and MG calculations were used to investigate the inter-action mechanism (fig. 5c). The calculated energy gap between HOMO and LUMO for ZIF-67 is 2.59 eV, whereas the calculated energy gap for MG is.121 eV. This difference indicates that MG requires more energy for the transition from LUMO to HOMO, whereas ZIF-67 requires less energy, indicating that it is more reactive. Their parameters are shown in Table-1.

The sections of a molecule with greater and lower potential are shown by the colours blue, yellow, and red in the MEP (Molecular Electrostatic Potential) study. Figure 7(d & eMEP )'s plots of ZIF-67 and MG indicate that the N atom of the imidazole ring is at a lower potential, which is shown by its reddish yellow hue, and may be a potential target for electrophilic assault. The H atoms in the imidazole ring have a lower potential, giving them a blue hue, and they may be vulnerable to nucleophilic assault. According to the MEP plot, the nitrogen of MG will experience nucleophilic contact, while the side chains of MG will experience electrophilic interaction. The results of the DFT study also support that due to the electrostatic interaction between the ZIF-67 and MG the adsorption process is spontaneous. [47]

  1. Poor format checking. Line 206: Table 2…since line 152 already stated it before.

 Response: Respected reviewer thank you very much for your valuable comments, we have removed all typo errors.

Overall, I would suggest major correction for the submitted manuscript.

Response: Respected reviewer thank you very much for your valuable time, we have incorporated all your valuable suggestions and now we think that this manuscript meets the acceptance requirements of “Molecules”.

Reviewer 2 Report

The manuscript reports the investigation of methyl green adsorption onto ZIF-67 in terms of sustainability, kinetics, thermodynamics and mechansim insights. The paper is well presented and can be accepted for publication after some minor corrections: 

1. It is recommended to include MG structural formula into the manuscript. 

2. Please compare your data and uptakes to other results published for the adsorption of methyl green onto different MOFs.

3. Please revise the manuscript for typos carefully, e.g. " Cu+2 + Kα radiation", "Co(NO3)2.H2O", "Co(NO3)3.6H2O", zinc mentioning (as a constituent of ZIF-67) instead of cobalt in last paragraphs, etc. 

4.What about recyclability of ZIF-67 in the adsorption of MG? 

Author Response

Dear and respected,

Ms. Claire Gao

Managing Editor,

Molecules.

Manuscript Ref. No.: MDPI: molecules-1999819

Article Type: Full-length article

Article Title: Facile synthesis of ZIF-67 for the adsorption of methyl green from wastewater: integrating molecular models and experi-mental evidence to comprehend the removal mechanism

Thank you very much for letting us revise our manuscript. No doubt the reviewer’s comments are valuable and very helpful for improving our research paper. We have studied the comments carefully and have tried our best to revise the manuscript, which we hope to meet with acceptance requirements. The reviewer’s comments have been yellow highlighted in the manuscript.

REVIEWER'S COMMENTS AND RESPONSES.

Comments and Suggestions for Authors

The manuscript reports the investigation of methyl green adsorption onto ZIF-67 in terms of sustainability, kinetics, thermodynamics and mechansim insights. The paper is well presented and can be accepted for publication after some minor corrections: 

  1. It is recommended to include MG structural formula into the manuscript. 

Response: Respected reviewer thank you very much for your valuable comments, we have added optimized structure of MG in figure 5a.

  1. Please compare your data and uptakes to other results published for the adsorption of methyl green onto different MOFs.

Response: Respected reviewer thank you very much for your valuable comments, we have added table-2 in the manuscript for comparison of MG removal by different MOFs. But according to the best of our knowledge only few papers have been published about the adsorptive removal of MG through MOFs. The adsorptive removal of MG with ZIF-67 has not been reported before in any literature.

  1. Please revise the manuscript for typos carefully, e.g. " Cu+2 +Kα radiation", "Co(NO3)2.H2O", "Co(NO3)6H2O", zinc mentioning (as a constituent of ZIF-67) instead of cobalt in last paragraphs, etc. 

Response: Respected reviewer thank you very much for your valuable comments, we have removed all typo errors and tried our best to polish our manuscript.

  1. What about recyclability of ZIF-67 in the adsorption of MG? 

Response: Respected reviewers thank you very much for your valuable comments, we re-used ZIF-67 four times and removal efficiency was decreased from 96 % to 94%.
